# Employee Motivation as a Tool to Achieve Sustainability of Business Processes

**Silvia Lorincová** [1], **Peter Štarchoň** [2], **Dagmar Weberová** [3], **Miloš Hitka** [1,*] and **Martina Lipoldová** [1]

1   Faculty of Wood Sciences and Technology, Technical University in Zvolen, Ul. T. G. Masaryka 24, 960 01 Zvolen, Slovakia

2   Faculty of Management, Comenius University in Bratislava, Odbojárov 10, P.O. BOX 95, 82005 Bratislava, Slovakia

3   Faculty of Multimedia Communications, Tomas Bata University in Zlín, Univerzitní 2431, 760 01 Zlín, Czech Republic

*   Correspondence: hitka@tuzvo.sk; Tel.: +421-45-520-6433

**Abstract:** Employee performance and their new ideas, as well as their efforts to promote the company in positive ways help build the values of an enterprise. Properly motivated managers, white-collar, and blue-collar workers use their performance to affect the business efficiency, and therefore the success and sustainability of the enterprise. Selecting the right structure of motivation factors, especially those aimed at job category and gender, is the main role of enterprise management. The aim of this study is to analyze and define differences in the perception of the preferred level of motivation in terms of gender and job category. The questionnaires were given to randomly selected employees working in Slovak enterprises in order to ensure variability and randomness of respondent selection which is necessary for relevant data acquisition. Following the average, the order of the importance of motivation factors of 3720 respondents was defined. The Student's t-test and Tukey's HSD test were used. We confirmed that there are statistically significant differences in the perception of the motivation in terms of job category. Moreover, we proved the significant differences between genders in the job category of blue-collar workers. We did not observe differences between genders in the other job categories studied. The results reported should be accepted and implemented in motivational programs by the employees of human resource departments as a way to keep up with strategic human resource management.

**Keywords:** strategic human resource management; sustainable work systems; employee motivation; job category; gender differences

## 1. Introduction

Sustainability in business process management is a key factor associated with the enterprise success [1,2]. Employees are considered one of the most important and leading factors in achieving sustainability [3–5], especially employees who move the business forward [6–9]. Managers, white-collar workers, and blue-collar workers are all involved in the company results. An employee's performance, their new ideas, as well as their efforts to promote the company in a positive way help build the values of an enterprise [10–13] and the success or failure of a business is affected by their productivity [14–21]. Employee productivity is influenced by employee motivation [22–25]. It is a complex and purposeful process to create a working environment and atmosphere that helps satisfy the aspirations, needs, and interests of employees and stimulates their action in a desirable way [26,27].

The quality of human potential plays an important role and it is a key factor that affects the running of a company, its prosperity, as well as sustainable development. A successful business is aware of the importance of its staff and their positive motivation; they are the greatest asset helping the company meet its goals. Currently, when advances in technology, information, and globalization occur most often, the human factor is becoming the biggest competitive advantage. The importance of human resources is considered strategic [28–32]. They become a part of strategic management of an enterprise and a factor important for sustainability. Effective employee management is supported by motivation.

A result assessment approach to employee management must focus on ways to encourage employee creativity, improve work performance, and create conditions that support team activity within the enterprise. It is connected with the employee performance in the workplace. Therefore, it is a specific task linked to the specific enterprise [33–37].

The motivation process is supported by setting realistic company goals and engaging employees. A motivational program focuses on the optimal use of the available workforce to meet company goals and, at the same time, on knowing and developing the personality of the employee. An effective motivational program covers the areas with low performance in a given period or those areas which seem to be significant for work activity due to another reason. The goal of the program is to create conditions encouraging motivation of all employees in the enterprise. Motivational programs affect employees in psychologically and economically ways, whereby the importance of both ways is equal. A motivational program is used especially as part of an adaptation programs. It is a document covering the set of facts affecting and motivating employees in accordance with the task relating to manufacturing, trade, and economic intentions of the enterprise [38].

We propose that motivation will be affected, besides other sociodemographic data (age, education, seniority, company strategic direction, region, and the size of an enterprise), by gender and job category. The aim of this study is to analyze and define differences in the perception of the preferred level of motivation in terms of gender and job category. The research is part of a long-term and extensive study on employee motivation in Slovakia dealing with the individual mentioned areas. In the future, the research results will be used to define the model of employee motivation in Slovak enterprises.

## 2. Literature Review

There is a wide range of tools used to motivate employees. F. Taylor defined money as the most important factor motivating employees to achieve higher productivity in industry [39,40]. This form of reward results in employee satisfaction and directly affects their performance. Salary is a valuable tool that plays an important role in the improvement of employee performance, as well as organizational productivity [41]. Studies [42–45] have shown that salary, promotion, bonuses, and other types of rewards are used by most enterprises to improve employee performance. Praise, setting realistic and achievable goals, appropriate workload definition, employee engagement, appropriate empowerment, responsibility, feedback, work equipment, expressing the positive personality features of a supervisor, appropriate leadership style, correctness by senior staff and company, and providing relevant information are considered to be other important motivation factors [46–54].

The role of business management is to define motivational factors that are used to manage and lead employees in an effective way. Current research has shown that the occurrence of differences in employee motivation depend on the employee's age [55–61]. However, in this process, the employee's position must be taken into account. With respect to the source of motivation for managers, they represent a specific group of employees [62]. Managers are motivated by financial motivational factors, as well as recognition and freedom in decision making [63,64]. Motivational factors for managers are often classified as "push" or "pull" factors. Push factors include the need to increase the family income, work dissatisfaction in terms of salary, difficulties finding a suitable job, and the need for flexibility due to family duties and responsibilities. Pull factors include the need for independence, self-actualization, and improvement of the current state and reputation in the society. White-collar

workers are motivated through rewards or recognition [65]. Employees at lower level job are also motivated by financial rewards [66,67].

When defining motivational factors, the role of enterprise management is to choose an appropriate structure of motivational factors with an emphasis on gender. Differences in motivation follow the differences in gender. Men put more effort into achieving wealth or financial well-being while women prefer work-life balance [68]. In general, women are motivated by family needs more than men whose priority is a private financial situation [69,70].

## 3. Materials and Methods

The level of employee motivation was investigated in this study conducted in 2018. The selection of respondents was proportionally allocated throughout Slovakia. All parts of Slovakia were covered by the research sample dataset. The questionnaires were given to randomly selected employees working in Slovak enterprises in order to ensure variability and randomness of respondent selection necessary for relevant data acquisition. A total of 3720 respondents, described in Table 1, participated in the research. Descriptive statistics were used to describe the primary sampling unit.

**Table 1.** Characteristics of respondents by job category.

| Job Category | Male | | Female | | Total | |
|---|---|---|---|---|---|---|
| | Absolute Frequency | Relative Frequency | Absolute Frequency | Relative Frequency | Absolute Frequency | Relative Frequency |
| Manager | 225 | 12.04 | 182 | 9.83 | 407 | 10.94 |
| White-collar worker | 588 | 31.46 | 1165 | 62.94 | 1753 | 47.12 |
| Blue-collar worker | 1056 | 56.50 | 504 | 27.23 | 1560 | 41.94 |
| Total | 1869 | 50.24 | 1851 | 49.76 | 3720 | 100.00 |

Source: Authors' compilation.

The following 30 motivational factors were examined: atmosphere in the workplace, good work team, fringe benefits, physical effort at work, job security, communication in the workplace, name of the company, opportunity to apply one's own ability, workload and type of work, information about performance result, working hours, work environment, job performance, career advancement, competences, prestige, supervisor's approach, individual decision making, self-actualization, social benefits, fair appraisal system, stress, mental effort, mission of the company, region's development, personal growth, relation to the environment, free time, recognition, and basic salary. Respondents assigned each motivational factor one of the five degrees of importance according to the Likert scale (5—very important, 4—important, 3—medium important, 2—slightly important, and 1—unimportant). The data gathered were processed using the STATISTICA 12 software. The importance of the level of motivation was investigated using the weighted arithmetic average formula. The level of motivation of all respondents was defined in terms of gender. Subsequently, the ten most important motivational factors for individual job categories of employees were defined. The motivational factors that were mentioned most occurred as the most important motivational factors over a long period in present studies [71–78]. A random variable, t, with Student t distribution was used as a test criterion for further testing. The following two hypotheses were tested at the level of significance $\alpha \leq 0.05$:

**Hypothesis 1.** *Statistically significant differences between genders are expected.*

**Hypothesis 2.** *Considering gender, statistically significant differences between job categories are expected.*

The likelihood of motivating employees, in terms of their gender and job category, with similar motivational programs was tested. The chi-Square or Pearson–Fisher ($\chi^2$) test was used to test the

agreement or disagreement between observations. Due to the selective character of the gathered data, Tukey's HSD (honest significant difference) at the significance level of 5% was used to test the differences between the averages of the values for the importance of motivational factors of white-collar workers. The Tukey's HSD test is a single-step multiple comparison procedure. It is modified for various numbers of observations in individual groups. Independence between levels of factors, variance, and normality agreement was expected.

## 4. Empirical Results

First, the dependence of motivational factors in terms of job category was verified. Tukey's HSD test was used. The results are presented in Figure 1.

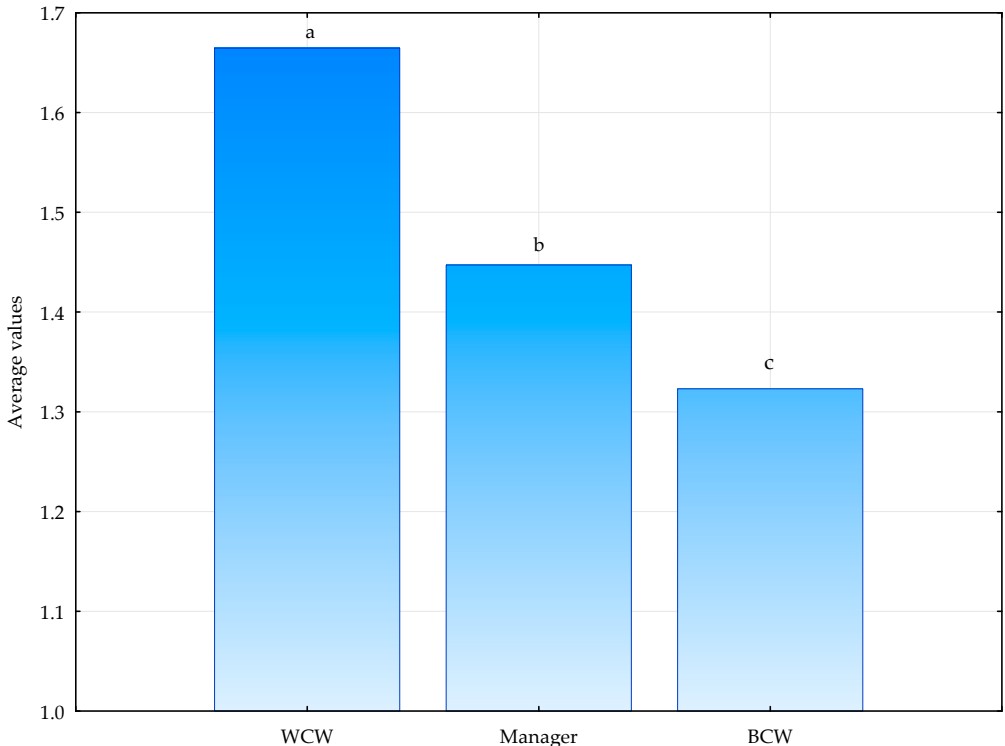

**Figure 1.** Testing the dependence of the average values between job categories. Note: WCW (white-collar worker), BCW (blue-collar worker).

The results in Figure 1 show that there were statistically significant differences in all job categories.

Subsequently, the importance of motivational factors in terms of gender was examined. The results are presented in Table 2.

The average values of 30 motivational factors in terms of gender are presented in Table 2. For men, the following 10 motivational factors were considered the most important: basic salary, atmosphere in the workplace, good work team, fringe benefits, fair appraisal system, supervisor's approach, job security, communication in the workplace, working hours, and work environment. For women, the motivational factors considered most important were: basic salary, atmosphere in the workplace, good work team, supervisor's approach, fair appraisal system, job security, fringe benefits, communication in the workplace, working hours, and work environment. The average values of these motivational factors were the highest rated.

When a detailed test at the level of $\alpha \leq 0.05$ was carried out, the occurrence of statistical dependence was confirmed for 17 out of 30 motivational factors. The statistically significant differences are highlighted in bold in Table 2. Following the results, the hypothesis, H1, was confirmed, i.e.,

there were statistically significant differences in the perception of the motivational level between men and women.

**Table 2.** Testing the dependence of the average values between genders.

| Motivational Factor | Male | Female | *p*-Level |
|---|---|---|---|
| Atmosphere in the workplace | 4.481 | 4.590 | 0.000015 *** |
| Good work team | 4.478 | 4.572 | 0.000864 *** |
| Fringe benefits | 4.414 | 4.407 | 0.014822 ** |
| Physical effort at work | 3.868 | 3.770 | 0.000003 *** |
| Job security | 4.375 | 4.441 | 0.074098 |
| Communication in the workplace | 4.299 | 4.377 | 0.015881 ** |
| Name of the company | 3.971 | 3.978 | 0.220068 |
| Opportunity to apply one's own ability | 4.044 | 4.082 | 0.121283 |
| Workload and type of work | 4.094 | 4.185 | 0.000284 *** |
| Information about performance result | 4.014 | 4.053 | 0.135037 |
| Working hours | 4.262 | 4.266 | 0.582611 |
| Work environment | 4.220 | 4.232 | 0.005689 ** |
| Job performance | 4.139 | 4.185 | 0.009677 ** |
| Career advancement | 4.060 | 4.025 | 0.042093 ** |
| Competences | 3.950 | 3.917 | 0.083130 |
| Prestige | 3.871 | 3.778 | 0.002051 ** |
| Supervisor's approach | 4.394 | 4.462 | 0.017969 ** |
| Individual decision-making | 4.014 | 4.050 | 0.014293 ** |
| Self-actualization | 4.017 | 4.055 | 0.065170 |
| Social benefits | 4.213 | 4.204 | 0.099271 |
| Fair appraisal system | 4.404 | 4.460 | 0.279517 |
| Stress | 4.089 | 4.206 | 0.000226 *** |
| Mental effort | 4.027 | 4.101 | 0.006000 ** |
| Mission of the company | 3.892 | 3.919 | 0.004789 ** |
| Region's development | 3.804 | 3.822 | 0.224187 |
| Personal growth | 4.056 | 4.083 | 0.025756 ** |
| Relation to the environment | 3.914 | 3.854 | 0.030853 ** |
| Free time | 4.137 | 4.096 | 0.346869 |
| Recognition | 4.163 | 4.213 | 0.031894 |
| Basic salary | 4.576 | 4.592 | 0.073355 |

Note: Single, double, and triple asterisks (*, **, ***) indicate significance at 5%, 1%, and 0.1% level. Source: Authors' compilation.

Furthermore, the importance of motivational factors in the case of job categories was examined in terms of gender.

### 4.1. The Level of Motivation in Terms of Job Category of the Manager

The job category, manager, was the first category analyzed. The results are presented in Table 3, indicating that the three most important motivational factors for men and women in the job category "manager" were the same. However, the order of importance was different. Male managers considered the basic salary the second most motivating factor, while, good work team was the second most important motivational factor for female managers.

The most important motivational factors for both men and women were chosen in order to test the dependence of motivational factors in terms of gender in the job category "manager". Following the Student t-test at the significance level $\alpha \leq 0.05$, statistically significant differences were not confirmed, i.e., there was no significant statistical dependence between selected motivational factors and gender in the job category "manager" (Table 4). On the basis of the results in the job category of manager, there was a high degree of similarity in motivational factors with a different order of preferences in motivational factors.

**Table 3.** Average values of selected motivational factors in the job category "manager".

| No. | Male | | Female | |
|---|---|---|---|---|
| | **Motivational Factor** | **Average** | **Motivational Factor** | **Average** |
| 1 | Atmosphere in the workplace | 4.569 | Atmosphere in the workplace | 4.654 |
| 2 | Basic salary | 4.560 | Good work team | 4.604 |
| 3 | Good work team | 4.542 | Basic salary | 4.604 |
| 4 | Fair appraisal system | 4.533 | Supervisor's approach | 4.533 |
| 5 | Supervisor's approach | 4.507 | Fair appraisal system | 4.522 |
| 6 | Job security | 4.476 | Job security | 4.484 |
| 7 | Communication in the workplace | 4.427 | Communication in the workplace | 4.478 |
| 8 | Fringe benefits | 4.413 | Individual decision making | 4.396 |
| 9 | Individual decision making | 4.369 | Fringe benefits | 4.385 |
| 10 | Personal growth | 4.369 | Selfactualization | 4.363 |

Note: Single, double, and triple asterisks (*, **, ***) indicate significance at 5%, 1%, and 0.1% level. Source: Authors' compilation.

**Table 4.** Testing the dependence of the most important motivational factors in terms of gender in the job category "manager".

| Motivation Factor | *p*-Level |
|---|---|
| Atmosphere in the workplace | 0.585 |
| Good work team | 0.332 |
| Fringe benefits | 0.066 |
| Job security | 0.879 |
| Communication in the workplace | 0.931 |
| Supervisor's approach | 0.373 |
| Individual decision making | 0.573 |
| Self-actualization | 0.334 |
| Fair appraisal system | 0.198 |
| Personal growth | 0.447 |
| Basic salary | 0.994 |

Source: Authors' compilation.

### 4.2. The Level of Motivation in Terms of Job Category of the White-Collar Worker

In the case of white-collar workers, basic salary, atmosphere in the workplace, and good work team were the three most important motivational factors for both men and women and the order of most importance factors was the same for both men and women. Further results are presented in Table 5.

**Table 5.** Average values of selected motivational factors in the job category "white-collar worker".

| No. | Male | | Female | |
|---|---|---|---|---|
| | **Motivational Factor** | **Average** | **Motivational Factor** | **Average** |
| 1 | Basic salary | 4.573 | Basic salary | 4.628 |
| 2 | Atmosphere in the workplace | 4.457 | Atmosphere in the workplace | 4.603 |
| 3 | Good work team | 4.457 | Good work team | 4.596 |
| 4 | Fringe benefits | 4.425 | Fair appraisal system | 4.493 |
| 5 | Supervisor's approach | 4.374 | Supervisor's approach | 4.481 |
| 6 | Fair appraisal system | 4.357 | Job security | 4.434 |
| 7 | Communication in the workplace | 4.320 | Fringe benefits | 4.426 |
| 8 | Job security | 4.316 | Communication in the workplace | 4.400 |
| 9 | Working hours | 4.219 | Working hours | 4.276 |
| 10 | Work environment | 4.204 | Recognition | 4.264 |

Source: Authors' compilation.

Statistically significant dependence between motivational factors and gender in the job category of white-collar workers was verified for selected motivational factors. The results in Table 6 show that there were statistically significant differences in selected motivational factors depending upon gender.

These factors included atmosphere in the workplace, good work team, job security, supervisor's approach, and fair appraisal system.

**Table 6.** Testing the dependence of the most important motivational factors in terms of gender in the job category "white-collar worker".

| Motivational Factor | *p*-level |
|---|---|
| Atmosphere in the workplace | 0.000 *** |
| Good work team | 0.001 *** |
| Fringe benefits | 0.226 |
| Job security | 0.025 ** |
| Communication in the workplace | 0.095 |
| Supervisor's approach | 0.004 ** |
| Working hours | 0.519 |
| Work environment | 0.694 |
| Fair appraisal system | 0.007 ** |
| Basic salary | 0.292 |

Note: Single, double, and triple asterisks (*, **, ***) indicate significance at the 5%, 1%, and 0.1% level. Source: Authors' compilation.

Testing the selected motivational factors with significant differences confirmed statistically are presented in Table 7.

**Table 7.** Testing the selected motivational factors in terms of gender in the job category "white-collar worker".

| Motivational Factor | Statistical Indicator | |
|---|---|---|
| Atmosphere in the workplace | Pearson's chi-square | 22.3240 |
| | Degree of freedom | df = 4 |
| | *p*-level | *p* = 0.000173 *** |
| Good work team | Pearson's chi-square | 18.3508 |
| | Degree of freedom | df = 4 |
| | *p*-level | *p* = 0.001054 ** |
| Job security | Pearson's chi-square | 11.1819 |
| | Degree of freedom | df = 4 |
| | *p*-level | *p* = 0.024594 ** |
| Supervisor's approach | Pearson's chi-square | 15.3366 |
| | Degree of freedom | df = 4 |
| | *p*-level | *p* = 0.004052 ** |
| Fair appraisal system | Pearson's chi-square | 13.9748 |
| | Degree of freedom | df = 4 |
| | *p*-level | *p* = 0.007376 ** |

Note: Single, double, and triple asterisks (*, **, ***) indicate significance at 5%, 1%, and 0.1% level. Source: Authors' compilation.

Five motivational factors with statistically significant differences and the overview of the values of importance assigned by respondents are shown in Table 8. Absolute and relative frequencies of responses are mentioned.

Selected motivational factors were considered important or very important by both men and women in the job category of white-collar workers. The value 5 (i.e., very important) was the value with the highest frequency of responses recorded in all motivational factors.

Average values, standard deviation, 95% confidence intervals in the primary sampling unit are mentioned in Table 9. Following the results presented in Table 9 the findings are generalized.

**Table 8.** The population proportion of individual score values of selected motivational factors in terms of gender in the job category "white-collar worker".

| Motivational Factor | Gender | Values of Importance | | | | | Total |
| | | 1 Unimportant | 2 Slightly Important | 3 Medium Important | 4 Important | 5 Very Important | |
|---|---|---|---|---|---|---|---|
| Atmosphere in the workplace | Male | 3 1% | 8 1% | 55 9% | 173 29% | 349 59% | 588 100% |
| | Female | 3 0% | 6 1% | 53 5% | 327 28% | 776 67% | 1165 100% |
| | Total | 6 | 14 | 108 | 500 | 1125 | 1753 |
| Good work team | Male | 3 1% | 7 1% | 41 7% | 204 35% | 333 57% | 588 100% |
| | Female | 1 0% | 6 1% | 49 4% | 351 30% | 758 65% | 1165 100% |
| | Total | 4 | 13 | 90 | 555 | 1091 | 1753 |
| Job security | Male | 5 1% | 12 2% | 77 13% | 192 33% | 302 51% | 588 100% |
| | Female | 10 1% | 12 1% | 107 9% | 369 32% | 667 57% | 1165 100% |
| | Total | 15 | 24 | 184 | 561 | 969 | 1753 |
| Supervisor's approach | Male | 1 0% | 12 2% | 53 9% | 222 38% | 300 51% | 588 100% |
| | Female | 7 1% | 18 2% | 82 7% | 359 31% | 699 60% | 1165 100% |
| | Total | 8 | 30 | 135 | 581 | 999 | 1753 |
| Fair appraisal system | Male | 6 1% | 19 3% | 57 10% | 183 31% | 323 55% | 588 100% |
| | Female | 12 1% | 18 2% | 86 7% | 317 27% | 732 63% | 1165 100% |
| | Total | 18 | 37 | 143 | 500 | 1055 | 1753 |

Source: Authors' compilation.

**Table 9.** Descriptive statistics and 95% confidence intervals for selected motivational factors in terms of gender in the job category "white-collar workers".

| Motivational Factor | Gender | N | Average | Standard Deviation | Confidence Interval | |
| | | | | | −95.00% | +95.00% |
|---|---|---|---|---|---|---|
| Atmosphere in the workplace | Male | 588 | 4.457 | 0.761 | 4.396 | 4.519 |
| | Female | 1165 | 4.603 | 0.627 | 4.567 | 4.639 |
| Good work team | Male | 588 | 4.457 | 0.722 | 4.399 | 4.516 |
| | Female | 1165 | 4.596 | 0.605 | 4.561 | 4.631 |
| Job security | Male | 588 | 4.316 | 0.839 | 4.248 | 4.384 |
| | Female | 1165 | 4.434 | 0.771 | 4.390 | 4.479 |
| Supervisor's approach | Male | 588 | 4.374 | 0.747 | 4.314 | 4.435 |
| | Female | 1165 | 4.481 | 0.745 | 4.438 | 4.524 |
| Fair appraisal system | Male | 588 | 4.357 | 0.861 | 4.287 | 4.427 |
| | Female | 1165 | 4.493 | 0.784 | 4.448 | 4.538 |

Source: Authors' compilation.

The results presented in Table 9 indicate that the motivational factor atmosphere in the workplace was assigned a value ranging from 4.396 to 4.519 by men in the job category of white-collar worker. Women in the same job category assigned the same motivational factor an average value in the range from 4.567 to 4.639 at the 95% confidence level. The results show that atmosphere in the workplace was evaluated in a more positive way by women than men in the job category of white-collar worker. Moreover, all analyzed motivational factors were rated higher by women in the job category of white-collar worker than men in the same job category.

Expected and residual frequencies of selected motivational factors in terms of gender in the job category of white-collar worker are presented in Table 10. Residual frequencies are the difference between frequencies in the line (discovered values in Table 8) and the expected frequencies of the evaluation of selected motivational factors.

**Table 10.** Expected and residual frequencies of selected motivational factors in terms of gender in the job category "white-collar worker".

| Motivational Factor | Frequency | Gender | Values of Importance | | | | |
|---|---|---|---|---|---|---|---|
| | | | 1 Unimportant | 2 Slightly Important | 3 Medium Important | 4 Important | 5 Very Important |
| Atmosphere in the workplace | Expected | Male | 2 | 5 | 36 | 168 | 377 |
| | | Female | 4 | 9 | 72 | 332 | 748 |
| | Residual | Male | 1 | 3 | **19** | 5 | −28 |
| | | Female | −1 | −3 | −19 | −5 | **28** |
| Good work team | Expected | Male | 1 | 4 | 30 | 186 | 366 |
| | | Female | 3 | 9 | 60 | 369 | 725 |
| | Residual | Male | 2 | 3 | 11 | **18** | −33 |
| | | Female | −2 | −3 | −11 | −18 | **33** |
| Job security | Expected | Male | 5 | 8 | 62 | 188 | 325 |
| | | Female | 10 | 16 | 122 | 373 | 644 |
| | Residual | Male | 0 | 4 | **15** | 4 | −23 |
| | | Female | 0 | −4 | −15 | −4 | **23** |
| Supervisor's approach | Expected | Male | 3 | 10 | 45 | 195 | 335 |
| | | Female | 5 | 20 | 90 | 386 | 664 |
| | Residual | Male | −2 | 2 | 8 | **27** | −35 |
| | | Female | 2 | −2 | −8 | −27 | **35** |
| Fair appraisal system | Expected | Male | 6 | 12 | 48 | 168 | 354 |
| | | Female | 12 | 25 | 95 | 332 | 701 |
| | Residual | Male | 0 | 7 | 9 | **15** | −31 |
| | | Female | 0 | −7 | −9 | −15 | **31** |

Source: Authors' compilation.

As shown in Table 10, atmosphere in the workplace tends to be evaluated by male white-collar workers as medium important, on the other hand, it is evaluated by female white-collar worker as very important. Moreover, men in the job category of white-collar worker, tend to rate analyzed motivational factors lower, with a lower degree of importance (medium important, important) than women in the same job category. Male white collar-workers tend to evaluate all analyzed motivational factors (atmosphere in the workplace, good work team, job security, supervisor's approach, and fair appraisal system) as very important.

*4.3. The Level of Motivation in Terms of Job Category of the Blue-Collar Worker*

The job category of the blue-collar worker was the third area studied. Basic salary was considered by male blue-collar workers as the most important motivational factor. On the other hand, female blue-collar workers considered atmosphere in the workplace the most important motivational factor. The importance of other motivational factors is presented in Table 11.

**Table 11.** Average values of selected motivational factors in the job category "blue-collar worker".

| No. | Male | | Female | |
|---|---|---|---|---|
| | Motivational Factor | Average | Motivational Factor | Average |
| 1 | Basic salary | 4.580 | Atmosphere in the workplace | 4.540 |
| 2 | Atmosphere in the workplace | 4.475 | Basic salary | 4.524 |
| 3 | Good work team | 4.475 | Good work team | 4.506 |
| 4 | Fringe benefits | 4.408 | Job security | 4.440 |
| 5 | Fair appraisal system | 4.403 | Supervisor's approach | 4.393 |
| 6 | Job security | 4.385 | Fringe benefits | 4.371 |
| 7 | Supervisor's approach | 4.382 | Fair appraisal system | 4.363 |
| 8 | Working hours | 4.267 | Communication in the workplace | 4.288 |
| 9 | Communication in the workplace | 4.259 | Working hours | 4.242 |
| 10 | Social benefits | 4.252 | Social benefits | 4.212 |

Source: Authors' compilation.

On the basis of the results of Student t-test shown in Table 12, we concluded that there were no statistically significant differences between the selected motivational factors and gender in terms of job category of the blue-collar worker. The research results in the job category of blue-collar worker show

that there was a high degree of similarity in motivational factors with different order preferences of motivational factors.

**Table 12.** Testing the dependence of the most important motivational factors in terms of gender in the job category "blue-collar worker".

| Motivational Factor | *p*-Level |
|---|---|
| Atmosphere in the workplace | 0.256 |
| Good work team | 0.609 |
| Fringe benefits | 0.139 |
| Job security | 0.604 |
| Communication in the workplace | 0.408 |
| Supervisor's approach | 0.351 |
| Working hours | 0.625 |
| Fair appraisal system | 0.339 |
| Social benefits | 0.651 |
| Basic salary | 0.117 |

Source: Authors' compilation.

## 5. Discussion

On the basis of the results of our research, we concluded that motivational factors such as basic salary, atmosphere in the workplace, as well as a good work team were highly motivating for all employees. However, men and women perceive the importance of these factors differently. Basic salary was a motivational factor of greater importance for men, whereas, women considered atmosphere in the workplace and a good work team more important. These findings correspond with the studies carried out in this field [68–70].

Further findings associated with the job category correspond with the research results of Bazydlo et al. [79] who showed that work environment, workplace comfort, and a good work team were the most motivating factors for managers. In Slovakia, employees with higher education are hired for manager positions. Their value orientation is due almost equally to their knowledge and gender equality [80–84]. In the case of managers, the results of our research show that a motivational program can be created regardless the gender and we did not observe any significant differences in motivational needs. The same conclusion was drawn in the case of blue-collar workers, especially when employees with primary and lower secondary education are hired for this job position. In addition, their value orientation is due almost equally to their knowledge and gender equality [80,85–87]. Following the analysis of motivation and education, similar results were observed.

In the case of white-collar workers, statistically significant differences in terms of gender were confirmed. Due to the statistically significant differences, the needs of individual groups had to be taken into account. Male white-collar workers tend to rate analyzed motivational factors lower as compared with women, who tend to evaluate analyzed motivational factors as very important.

There were statistically significant differences in perception of motivation among the three job categories mentioned in Figure 1. Therefore, a different motivational program must be created for each job category.

Furthermore, our research results indicate that blue-collar workers were motivated by the amount of money they receive in the form of basic salary. This was confirmed by other studies [66,67,88,89].

In general, the fact that there were statistically significant differences in motivation between men and women is considered the main finding. In terms of job categories of managers and blue-collar workers, motivational programs can be created regardless of gender. In the case of white-collar workers, motivational program must vary due to gender.

## 6. Conclusions

The statement that quality human resources have become an integral part of the company's strategy has been confirmed by [90,91]. Employees play a key role in the implementation of the overall business development strategy. The efficiency of business processes, and therefore the overall success of the enterprise is affected by the performance of properly motivated employees [92–97]. Results of our research show that there were statistically significant differences in perceiving the motivation in terms of gender. In the case of mixed employee teams, this fact must be taken into consideration in the process of designing motivational programs. Despite the similarity in the order of the importance of motivational factors in terms of men and women, both of them perceived the individual motivational factors in different ways.

The aim of this study was to define the differences in the perception of the level of motivation in terms of gender and job category. The fact that there are statistically significant differences in the perception of motivation in terms of job category was proven. The significant differences in the job category of blue-collar workers in terms of gender were proven as well. In the case of two other job categories, no significant differences between genders were observed. The fact that the aim of the study was met can be stated. The results should be accepted and implemented in motivational programs by the employees of the human resource department. In the future, we plan to find correlations between other sociodemographic data (age, education, seniority, company strategic direction, region, the size of an enterprise) and use our results to define a model for employee motivation in enterprises. However, further data collection and analysis is required.

**Author Contributions:** Conceptualization, S.L., P.Š., D.W., and M.H.; methodology, S.L., P.Š., D.W., M.H., and M.L.; data curation, S.L., P.Š., D.W., M.H., and M.L.; writing—original draft, S.L., P.Š., D.W., and M.H.; visualization, S.L., P.Š., D.W., and M.H.

**Funding:** This research was funded by VEGA 1/0024/17 "Computational model of motivation", and APVV 16-0297 "Updating of anthropometric database of Slovak population".

**Conflicts of Interest:** The authors declare no conflict of interest.

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
