# Peer review of "Employee Motivation as a Tool to Achieve Sustainability of Business Processes"

_sustainability, doi:10.3390/su11133509_

Round 1

Reviewer 1 Report

The data appears to be very good, but the analysis could be much expanded and improved. There was special significance assigned to the top 10 areas of both male and female respondents, but there is nothing to support why 10 were chosen other than they are the top 10. Was there some threshold for significance for these top 10, or was it just a round number and the top 10 were chosen?

Furthermore, on face value, there does not seem to be much difference between the responses for the categories themselves. Some sort of multivariable regression should be used such as Ordinal Logistic Regression or ordered logit, but a statistician should be consulted.

Don't include things such as the formulas to calculate the average or t-test. Anyone reading the paper would know this.

Don't highlight with bold type. It does not show up very well, use asterisks and define what level of statistical significance they represent. You can use more asterisks to show higher levels of significance if desired. Look at other papers to see how this is done.

Use a bar graph to show the level of importance for each response type and use letters on the respective bars to represent when Tuckeys HSD shows a statistical difference. This is much more visual than trying to look at all the numbers in a chart. Look at other papers to see how this is done.

I have attached some examples.

Author Response

Dear Reviewer,

thank you for your review. The recommendations were incorporated. All the changes provided are highlighted.

Comments and Suggestions for Authors

The data appears to be very good, but the analysis could be much expanded and improved. There was special significance assigned to the top 10 areas of both male and female respondents, but there is nothing to support why 10 were chosen other than they are the top 10. Was there some threshold for significance for these top 10, or was it just a round number and the top 10 were chosen?

The level of motivation of all respondents was defined in terms of gender. Subsequently, ten most important motivation factors for individual job categories of employees were defined. Most mentioned motivation factors have occurred as the most important motivation factors over a long period in our present research studies (Lorincová et al., 2018; Hitka et al., 2017; Kampf, et al., 2017; Lorincová et al., 2016; Myšková et al., 2016; Hitka et al., 2015; Kampf et al., 2014; Hitka et al., 2011).

Furthermore, on face value, there does not seem to be much difference between the responses for the categories themselves. Some sort of multivariable regression should be used such as Ordinal Logistic Regression or ordered logit, but a statistician should be consulted.

Based on our previous research (Lorincová et al., 2018; Hitka et al., 2017; Kampf, et al., 2017; Lorincová et al., 2016; Myšková et al., 2016; Hitka et al., 2015; Kampf et al., 2014; Hitka et al., 2011), we can state that creating a motivational program from a large number of motivation factors would be highly costly and time consuming. For this reason, we focus on the most important motivation factors that were obtained by an average. We consider your recommendation highly interesting, and we will try to deal with this area in our future research. However, we think that this may cause a great extension of the article, which would not correspond to the requirements of the Journal.

Don't include things such as the formulas to calculate the average or t-test. Anyone reading the paper would know this.

Formulas were deleted.

Don't highlight with bold type. It does not show up very well, use asterisks and define what level of statistical significance they represent. You can use more asterisks to show higher levels of significance if desired. Look at other papers to see how this is done.

The recommendation was accepted and asterisks were used.

Use a bar graph to show the level of importance for each response type and use letters on the respective bars to represent when Tuckeys HSD shows a statistical difference. This is much more visual than trying to look at all the numbers in a chart. Look at other papers to see how this is done. I have attached some examples.

The figure 1 was added.

Reviewer 2 Report

I would like to thank the authors for the effort in writing this research paper. The topic of motivation although widely studied and examined, remains very attractive and a field of study quite rich. What motivates employees with different characteristics is a major concern of practitioners.

Although the topic is relevant, I would like to share some ideas and make some comments regarding the structure and content of your paper. Please consider this observations as suggestions for improvement.

The abstract does not follow an appropriate structure. It is not clear what’s the research goal, the methods used and the main findings

You choose not to separate the literature review from the introduction. Although this a viable option, it is important that the usual topics of introduction (e.g. the subject under study, brief history of the subject, research problem, the research objective) and literature review (e.g. which is the theoretical option, models and/or theory) are present. 

In the “materials and methods” section, although you have an impressive sample size, you do not describe your population nor explain how data was collected. Also, you do not explain why you use those specific 30 motivating factors, and the source(s) you based on.

In lines 94 and 96 the assumptions are the same although the formula is different.

You choose not to separate the discussion from the conclusions. Again, this is a viable option, but it’s important to approach some topics. For example you do not offer a proper discussion of the findings, limiting your discussion to the agreement of the findings with the literature. As an example, what do you think might explain the lack of differences between managers and blue-collar workers? What the literature as to offer about this topic? (you even have a reference with a compelling title about this topic: What motivates employees? Workers and supervisors give different answers)

Some important information that usually is mentioned on the conclusion is also missing. For example, resume the objectives and make it clear if they have been achieve, explicitly indicate the contributions of the paper, inform the limitations of your research, and suggest future studies (if possible).

Finally, although you have an impressive list of references, it’s hard to understand the relevance of most of them for the topic being studied (e.g. Gejdoš, M.; Tončíková, Z.; Němec, M.; Chovan, M.; Gergeľ, T. Balcony cultivator: New biomimicry design approach in the sustainable device. Futures 2018, 98, 32-40. doi: 10.1016/j.futures.2017.12.008 281 OR Nedeliaková, E.; Štefancová, V.; Kuka, A. Quick Response quality control as an innovative approach in the conditions of rail transport. In Proceedings of the 12th International Conference Quality Production 283 Improvement, QPI 2018, Zaborze near Myszkow, Poland, 18–20 June 2018 284 OR Ližbetin, J.; Vejs, P.; Stopka, O.; Cempírek, V. The significance of dynamic detection of the railway vehicles weight. Nase More 2016, 63, 156–160. 334, but there are more)

You should not use references from thesis or dissertations (e.g. Hanzl, J. A Proposal of Alternative Routes to Highways in Case of Extraordinary or Planned Traffic Restrictions: Dissertation; Czech Technical University in Prague: Prague, Czech Republic, 2017. 288)

Author Response

Dear Reviewer,

thank you for your review. The recommendations were incorporated. All the changes provided are highlighted.

Comments and Suggestions for Authors

I would like to thank the authors for the effort in writing this research paper. The topic of motivation although widely studied and examined, remains very attractive and a field of study quite rich. What motivates employees with different characteristics is a major concern of practitioners.

Although the topic is relevant, I would like to share some ideas and make some comments regarding the structure and content of your paper. Please consider this observations as suggestions for improvement.

The abstract does not follow an appropriate structure. It is not clear what’s the research goal, the methods used and the main findings.

The abstract was modified. Aim, methods and main findings were added.

You choose not to separate the literature review from the introduction. Although this a viable option, it is important that the usual topics of introduction (e.g. the subject under study, brief history of the subject, research problem, the research objective) and literature review (e.g. which is the theoretical option, models and/or theory) are present. 

Introduction and literature review was separated. New references were added.

In the “materials and methods” section, although you have an impressive sample size, you do not describe your population nor explain how data was collected. Also, you do not explain why you use those specific 30 motivating factors, and the source(s) you based on.

Our research is part of a long-term and extensive research of the employee motivation of employees working in Slovak enterprises where we focus on the individual areas mentioned above (Lorincová et al., 2018; Hitka et al., 2017; Kampf, et al., 2017; Lorincová et al., 2016; Myšková et al., 2016; Hitka et al., 2015; Kampf et al., 2014; Hitka et al., 2011).

In lines 94 and 96 the assumptions are the same although the formula is different.

Formulas were deleted.

You choose not to separate the discussion from the conclusions. Again, this is a viable option, but it’s important to approach some topics. For example you do not offer a proper discussion of the findings, limiting your discussion to the agreement of the findings with the literature. As an example, what do you think might explain the lack of differences between managers and blue-collar workers? What the literature as to offer about this topic? (you even have a reference with a compelling title about this topic: What motivates employees? Workers and supervisors give different answers). Some important information that usually is mentioned on the conclusion is also missing. For example, resume the objectives and make it clear if they have been achieve, explicitly indicate the contributions of the paper, inform the limitations of your research, and suggest future studies (if possible).

The discussion and the conclusions were separated. More findings and limitations were added.

Finally, although you have an impressive list of references, it’s hard to understand the relevance of most of them for the topic being studied (e.g. Gejdoš, M.; Tončíková, Z.; Němec, M.; Chovan, M.; Gergeľ, T. Balcony cultivator: New biomimicry design approach in the sustainable device. Futures 2018, 98, 32-40. doi: 10.1016/j.futures.2017.12.008 281 OR Nedeliaková, E.; Štefancová, V.; Kuka, A. Quick Response quality control as an innovative approach in the conditions of rail transport. In Proceedings of the 12th International Conference Quality Production 283 Improvement, QPI 2018, Zaborze near Myszkow, Poland, 18–20 June 2018 284 OR Ližbetin, J.; Vejs, P.; Stopka, O.; Cempírek, V. The significance of dynamic detection of the railway vehicles weight. Nase More 2016, 63, 156–160. 334, but there are more)

You should not use references from thesis or dissertations (e.g. Hanzl, J. A Proposal of Alternative Routes to Highways in Case of Extraordinary or Planned Traffic Restrictions: Dissertation; Czech Technical University in Prague: Prague, Czech Republic, 2017. 288)

List or references was redefined.

Round 2

Reviewer 2 Report

Dear Authors

Thank you for taking the time to review the paper.

Generally, the topics I have mention were addressed.

There are still minor concerns (for example, you did not addressed properly the comments about the methods), but less relevant for the general quality of the paper.